# Effects of *Pseudomonas* sp. OBA 2.4.1 on Growth and Tolerance to Cadmium Stress in *Pisum sativum* L.

**DOI:** 10.3390/biotech12010005

**Published:** 2023-01-03

**Authors:** Liliya Khakimova, Olga Chubukova, Zilya Vershinina, Dilara Maslennikova

**Affiliations:** 1Institute of Biochemistry and Genetics—Subdivision of the Ufa Federal Research Centre of the Russian Academy of Sciences, 450054 Ufa, Russia; 2Department of Fundamental and Applied Microbiology, Bashkir State Medical University (BSMU), 450008 Ufa, Russia; 3Department of Molecular Technologis, Federal State Budgetary Educational Institution of Higher Education “Ufa State Petroleum Technological University” (USPTU), 450000 Ufa, Russia

**Keywords:** *Pseudomonas* sp., PGPR, heavy metals (HMs), cadmium tolerance, *Pisum sativum* L., biofilms, growth stimulation, malondialdehyde

## Abstract

Cadmium stress is a barrier to crop production, yield, quality, and sustainable agriculture. In the current study, we investigated the characteristics of bacterial strain *Pseudomonas* sp. OBA 2.4.1 under cadmium (CdCl_2_) stress and its influence on Cd stresses in pea (*Pisum sativum* L.) seedlings. It was revealed that strain OBA 2.4.1 is tolerant of up to 2 mM CdCl_2_, and seed treatment with the bacterium enhanced pea plant growth (length of seedlings) under 0.5 mM cadmium stress. This bacterial strain showed plant growth-promoting properties, including biofilm formation and siderophore activity. An important advantage of the studied strain OBA 2.4.1 is its ability to colonize the plant roots. Moreover, the inoculation with strain OBA 2.4.1 significantly reduced oxidative stress markers in pea seedlings under cadmium stress. These findings suggest that cadmium stress-tolerant strain OBA 2.4.1 could enhance pea plant growth by mitigating stress-caused damage, possibly providing a baseline and eco-friendly approach to address heavy metal stress for sustainable agriculture.

## 1. Introduction

Biotic and abiotic stresses affect plant growth, development, and crop yields. Heavy metals (HMs) adversely affect all processes occurring in plants and, in addition, are very toxic to the human body and can cause serious health problems for humans [1,2]. The rapid development of industry and manufacturing causes serious environmental pollution, especially in soil. HMs in the soil are absorbed by plant roots and accumulate in all plant tissues, seriously slowing down many physiological and molecular processes [3]. Excessive accumulation of Cd^2+^ in plants leads to growth retardation, a decrease in the contents of chlorophyll and carotenoids, as well as a decline in leaf area and photosynthesis rate, along with reduced plant biomass and water content and increased protease activity [4]. Cd^2+^ in plants can bind to proteins, causing denaturation and dysfunction, resulting in growth inhibition [5]. Cd can lead to the formation of reactive oxygen species (ROS) [1,6]. Sandalio et al. (2001) showed that Cd induces oxidative stress in *Pisum sativum* seedlings, promoting the accumulation of lipid peroxide and oxidized protein, and reducing catalase and superoxide dismutase (SOD) activity [7].

To mitigate the problems associated with HMs contamination, there is an urgent need for alternative environmentally friendly technologies, such as the use of plant growth-promoting microorganisms/rhizobacteria (PGPM/PGPR), for future use. This includes various taxonomic groups that have a wide range of beneficial properties for plants. Rhizospheric bacteria *Pseudomonas* sp. Belongs to PGPM [8,9]. These microorganisms actively colonize the plant roots, stimulate their growth, activate the immune response, and suppress the growth of soil phytopathogenic fungi and bacteria. Unlike other soil bacteria, *Pseudomonas* sp. Grows very quickly [8,9]. Currently, *P. fluorescens, P. aureofaciens, P. chlororaphis, P. corrugate, P. putida*, and others strains can be used as growth-stimulating agents, as well as protectors from the negative effects of HMs [8,9,10]. 

Phytoremediation by PGPR is an effective environmental measure to increase the remediation of HMs-contaminated soils. There are works showing the positive effect of Cd-Zn-tolerant PGPR *Bacillus* sp. Strain ZC3-2-1 in *Oryza sativa*. Treatment with this strain significantly increased *O. sativa* biomass, but the content of Cd^2+^ and Zn^2+^ per unit of rice biomass did not change significantly. This fact proves that the *Bacillus* sp. ZC3-2-1 can increase the efficiency of the phytoremediation of soils contaminated with Cd-Zn, promoting phytoextraction and metal immobilization [11].

Plants of the *Brassicaceae* family are most often recommended as phytoremediation plants, since they have a relatively high resistance to HMs and exhibit the ability to accumulate them [12,13,14]. However, their main disadvantage in phytoremediation applications is their low growth rate and low biomass yield. As an alternative, it is possible to consider plants of the *Fabaceae* family, which grow rapidly and gain a large biomass, but have a relatively low tolerance to HMs [15,16]. In addition, legumes form a nitrogen-fixing symbiosis with nodule bacteria and an associative symbiosis with PGPR, exerting a positive effect on plant growth and nutrition [17]. Such symbiotic interactions of legumes with beneficial bacteria increase plant adaptation to stresses, which makes it possible to use them for the bioremediation of polluted soils [9,17,18,19,20,21].

Previously, when studying the growth-promoting properties of *Pseudomonas* spp., four strains were selected: OBA 2.4.1, OBA 2.9, GOR 4.17, and STA 3, which were isolated from the rhizosphere of plants in the Southern Urals (Russia). The most effective strain of *Pseudomonas* sp. Was chosen for further research, particularly, the strain OBA 2.4.1, which increased the germination vigor of *Medicago sativa* L. seeds by 15% compared to the untreated control. In addition, among the studied strains of *Pseudomonas* sp., OBA 2.4.1 showed the greatest resistance to growth on media under Cd stress [22]. Therefore, the aim of the present study was to determine the different activity and inhibitory concentrations of Cd salts toward the growth of *Pseudomonas* sp. OBA 2.4.1, analyzing its effect on *Pisum sativum* L. growth and tolerance under Cd stress. 

## 2. Materials and Methods

### 2.1. Isolation of Pseudomonas sp. OBA 2.4.1 

The strain of *Pseudomonas* sp. OBA 2.4.1 was isolated from the rhizosphere of the *Oxytropis baschkiriensis* by homogenizing soil samples in a sterile LB medium (Bacto Tryptone, 1%; yeast extract, 0.55; NaCl, 0.5%; agar, 1%), with further subculture and growth at 28°C [23]. DNA from the bacteria was isolated by cell lysis in 1% Triton X100 and 1% Chelex100 suspension [24].

### 2.2. Molecular Genetic Identification of Bacterial Strains

To identify the isolated bacteria, the 16S rRNA gene was amplified using the universal primers fD1 5’- CCCGGGATCCAAGCTTAAGGAGGTGATCCAGCC-3’ and rD1 5’- CCGAATTCGTCGACAACAGAGTTTGATCCTGGCTCAG -3’ [25]. For amplification of the rpoD gene fragment, the primers PsEG30F5’-ATYGAAATCGCCAARCG-3’ and PsEG790R5’-CGGTTGATKTCCTTGA-3’ were used [26,27].

Nucleotide sequences were determined on an Applied Biosystems 3500 automatic sequencer (Applied Biosystems, Waltham, MA, USA) using the Big Dye Terminator v. 3.1. The analysis was carried out using the Lasergene software package (DNASTAR, Inc., Madison, WI, USA). Nucleotide sequences for comparative analysis were taken from the GenBank database (www.ncbi.nlm.nih.gov accessed on 28 December 2022). Computational analysis of DNA sequence fragments was performed using the Clustal W multiple alignment method in the Megalign Lasergene program (DNASTAR, Inc. Madison, WI, USA).

To obtain a fluorescently labeled strain of *Pseudomonas* sp. OBA 2.4.1, the pJNTurboRFP vector was used [28]. Some of the root fragments were used to visually assess the colonization of the surface of plant root hairs by bacteria using an Axio Imager M1 fluorescent microscope (Carl Zeiss, Jena, Germany). For this, the roots were incubated together with labeled bacteria (10^6^ CFU/mL) for 1 day.

### 2.3. Biofilm Formation on Inert Surfaces

Biofilms were examined on 24-well plastic plates (polystyrene) (Corning, Inc., USA). Bacteria were grown for 24 h in Lauria–Bertani (LB) medium at 28°C and 140 rpm. The bacterial culture was diluted to 10^6^ CFU/mL and 1 mL was transferred into the wells of the plate and incubated at 28°C and 50 rpm for 7 days. To determine the relative biofilm density, the gentian violet staining method (Agat-Med, Russia) was used [29]. The optical density of the samples was measured using an Enspire Model 2300 Multilabel Microplate Reader (Perkin Elmer, Boston, MA, USA).

### 2.4. Phosphate Mobilization and Siderophore Activity of Pseudomonas sp. OBA 2.4.1 

The determination of the ability of the *Pseudomonas* sp. OBA 2.4.1 to mobilize inorganic phosphorus was tested on plates with Muromtsev’s medium (glucose 10 g/L, asparagine 1 g/L, K_2_SO_4_ 0.2 g/L, MgSO_4_ 0.2 g/L, corn extract 0.02 g/L, agar 20 g/L; pH 6.8) containing insoluble phosphate. As a source of phosphorus, Ca_3_(PO_4_)_2_ was added to the medium at a concentration of 5 g/L [30]. A daily culture of bacteria was applied as a drop on the surface of an agar medium and incubated at a temperature of 28°C.

CAS-blue agar was prepared as described by Schwyn (1986), with some modifications [31,32]. To prepare 100 mL of the medium, 6.5 g of chrome azurol S was dissolved in 5 mL of water and mixed with 1 mL of a solution containing 1 mM FeCl_3_ and 10 mM HCl. After that, 4 mL of a solution containing 7.3 mg of HDTMA was added to the chrome azurol solution. The resulting mixture was autoclaved and added to the sterile LB medium. The bacteria were grown on the medium for 2 days. A change in color to yellow, orange, or pink revealed the release of siderophores.

### 2.5. Determination of H_2_O_2_ Content

Hydrogen peroxide (H_2_O_2_) levels were determined according to [33]. The samples of plant material were homogenized (1:5 weight/volume) in 0.05M sodium phosphate buffer pH 6.2. The supernatant was separated by centrifugation (Eppendorf^®^ Microcentrifuge 5415 R, Humburg, Germany, USA) at 15,000× g for 15 min. The concentration of H_2_O_2_ in the supernatant was spectrophotometrically (SmartSpecTM Plus, Bio–Rad, Hercules, CA, USA) determined using xylenol orange in the presence of Fe^2+^ at 560 nm. The H_2_O_2_ was expressed as µmoL g^–1^ FW.

### 2.6. Determination of the Malondialdehyde (MDA) Content

For the measurement of lipid peroxidation in leaves, the thiobarbituric acid (TBA) test, which determines MDA as an end product of lipid peroxidation [34], was used. Plant material were ground in distilled water and then homogenized in 20% trichloroacetic acid (TCA). The homogenate was centrifuged at 10,000× *g* for 20 min, and 0.5 mL of the supernatant was added to 1 mL 0.5% (w/v) TBA in 20% TCA. The mixture was incubated in boiling water for 30 min, and the reaction was stopped by placing the reaction tubes in an ice bath. Then, the samples were centrifuged at 10,000× *g* for 5 min. Absorbance was spectrophotometrically (SmartSpecTM Plus, Bio–Rad, Hercules, CA, USA) measured at 532 nm. The amount of MDA–TBA complex (red pigment) was calculated from the extinction coefficient 155 mM^−1^ cm ^−1^. The MDA was expressed as nmoL g^–1^ FW.

### 2.7. Treatment of Plants with Pseudomonas sp. OBA 2.4.1 

Pea seeds (*Pisum sativum* L., Kelvedonskoye miracle variety) were sterilized in 70% ethyl alcohol for 1 min and 10% sodium hypochlorite solution for 20 min [29]. Thereafter, the seeds were treated with *Pseudomonas* sp. OBA 2.4.1 (10^7^ CFU/mL) for 30 min and germinated on filter paper with sterile water (control) and various concentrations of CdCl_2_ (0.1, 0.2, 0.3, 0.4, and 0.5 mM) (stress) at 24 ± 1°C in the dark for 7 days. Seven-day-old seedlings were taken to assess their shoot length. For the analysis, 50 seedlings were used in each variant in three independent biological replicates.

### 2.8. Statistical Analysis 

All microbiological, molecular, biochemical, and physiological experiments were performed in three or more bioassays and three or four analytical tests. The arithmetic average values and confidence intervals calculated from the standard error are shown in the table and graphs (± SEM). Statistically significant differences between the mean values were evaluated using analysis of variance (ANOVA), followed by the Tukey test (p < 0.05).

## 3. Results

### 3.1. Identification of the Strain OBA 2.4.1 and Its Main PGP Traits

The sequenced 16S rRNA sequence was deposited in GenBank (http://www.ncbi.nlm.nih.gov/genbank accessed on 28 December 2022) under the number OK039351 and in the rpoD gene under the number OM641958. When comparing fragments of the 16S rRNA and rpoD gene sequences with typical strains, it was found that *Pseudomonas* sp. OBA 2.4.1 is closest in homology to *Pseudomonas fluorescens*. This strain is described in detail in a previous study [27].

It was revealed that the strain *Pseudomonas* sp. OBA 2.4.1 is capable of forming biofilms on inert surfaces. In addition, siderophore activity was also observed (Figure 1). 

### 3.2. Growth Analysis of Pseudomonas sp. OBA 2.4.1 under Cadmium Stress (0.5, 1, 1.5, 2 mM CdCl_2_)

The presence of Cd in the growth medium of the bacteria led to the inhibition of the growth of their colonies, especially in the presence of 1.5. and 2 mM Cd (Figure 2A). The growth analysis of the labelled bacterial colonies showed a similar result (Figure 2B).

### 3.3. Effect of Pseudomonas sp. OBA 2.4.1 Strain on Pea Growth under Cd Stress

It was found that the morphometric parameters showed significant visual differences (Figure 3). Cd stress inhibits the germination and growth of pea plants (Figure 3A). The most negative effect on growth was exerted by 0.5 mM Cd. Seed pretreatment with bacterial strain OBA 2.4.1 contributed to the reduction of the inhibitory effect of Cd,. Under the action of 0.1 mM Cd, the length of these plants increased by 62%, of 0.2 mM by 20%, of 0.3 mM by 36%, of 0.4 mM by 51%, and of 0.5 mM by 118% relative to the control stressed plants (Figure 3B). 

It was found that treatment with *Pseudomonas* sp. OBA 2.4.1 stimulated the growth of pea seeds; in the presence of Cd, a gradual deterioration in the growth and development of seedlings was also observed (Table 1). 

### 3.4. Effects of Strain Pseudomonas sp. OBA 2.4.1 on the Content H_2_O_2_ and MDA in Pea Seedlings under Cd Stress

The results showed that Cd stress resulted in a more than twofold increase in H_2_O_2_ content (Figure 4).

This result is accompanied by the same level of MDA accumulation (Figure 5), but treatment with *Pseudomonas* sp. OBA 2.4.1 significantly reduced the damage caused by stress. Thus, the content of H_2_O_2_ and MDA the outflow was higher (relative to the control) by 1.5 times and 1.6 times, respectively. At the same time, the treatment itself with *Pseudomonas* sp. OBA 2.4.1 did not affect the state of the membrane structures under normal conditions.

### 3.5. Bacterial Colonization of Pisum sativum L. Plant Roots

Figure 6 shows the growth of *Pseudomonas* sp. OBA 2.4.1 transformed with the pJNTurboRFP vector. The bacteria have not lost their resistance to Cd. The fluorescent label made it possible to visualize bacterial cells on the surface of the root hairs of germinated pea seedlings. Microscopy showed the formation of bacterial clusters and microcolonies of *Pseudomonas* sp. OBA 2.4.1 on the roots of peas.

## 4. Discussion

For successful colonization of plant roots, the bacteria must be highly competitive. For example, the ability to form biofilms has a positive effect on the survival strategy of bacteria. There are studies showing that when *P. aeruginosa* bacteria formed a biofilm, they became more resistant to HMs ions (Cu^2+,^ Pb^2+^, Zn^2+^) compared to single bacteria, since polymer compounds bind metal ions, preventing them from entering the biofilm [34,35]. The studied strain of *Pseudomonas* sp. OBA 2.4.1 also has the ability to form a biofilm, which characterizes it as a PGPM with good competitiveness. Another indicator of PGPM quality was the discovery of siderophore activity (Figure 1), since phosphate mobilization and siderophore activity are also traits of PGPM. Phosphate mobilization helps plants absorb phosphorus, and the secreted siderophores inhibit the growth of pathogenic fungi by reducing the amount of iron available to them in the soil. The ability to produce siderophores is another important feature of PGPM involved in plant growth stimulation [36,37]. Siderophores are low molecular weight (<1000 Da) molecules with a high specificity and affinity for a chelate or Fe^3+^ bond. They play an important role in stimulating plant growth, increasing resistance, and protecting against pathogens [38]. 

OBA 2.4.1 showed good growth on medium with cadmium stress at concentrations up to 1 mM (Figure 2A). Further, with an increase in Cd, the growth of bacteria was markedly inhibited. Similar growth was also observed in labeled bacteria (Figure 2B). This fact shows that labeled bacteria do not lose their resistance to cadmium stress.

It is well known that the antioxidant system plays a fundamental role in maintaining the redox homeostasis of plants under stress [1]. As expected, the presence of Cd in the growth medium led to the development of oxidative stress [1,39], which was accompanied by the depletion of glutathione (GSH) and ascorbate (AsA) pools, as well as the stress-induced activation of GR and APX. The over-accumulation of ROS led to the excessive synthesis of MDA and an increase in the permeability of the membrane structures. An excess of MDA, the end product of lipid peroxidation, and a depletion of GSH, which is a fundamental molecule regulating mitosis [33,39], led to the inhibition of plant growth under stress [32].

An important indicator for assessing the prospects for the use of bacteria is the assessment their influence on plant growth parameters. In this work, it was found that the strain *Pseudomonas* sp. OBA 2.4.1 stimulated pea seed growth (Figure 3). The data obtained showed that inoculation of pea seeds with OBA 2.4.1 had a positive effect on the length of the seedlings, which may indicate an increase in plant resistance to cadmium stress at the initial stage of plant growth. Using labeled bacteria, we visually confirmed that they colonize seedling root hairs well. Due to their properties, bacteria have a beneficial effect on growth under cadmium stress.

When growing seeds under cadmium stress at low concentrations of Cd (0.1, 0.2 mM), an improvement in the growth of seedlings was observed. Further, an increase in the concentration of Cd led to the inhibition of seedlings growth. A similar effect is observed in many studies regarding cadmium stress. For example, Jalil et al. (1994) found that a low concentration of Cd^2+^ can promote the growth of durum wheat, but at the same time, at a relatively high concentration, wheat growth was inhibited, and Cd resistance varied in different varieties [40]. Yang et al. (2005) found that when the concentration of Cd^2+^ was 0-1 mg/kg, the height of the vine also increased, indicating a beneficial effect of Cd^2+^ on vine growth [41]. It is also worth noting that with an increase in the growth period under cadmium stress, plant growth is also inhibited; as shown in the study by Liu (2004), the height of corn seedlings under Cd^2+^ treatment is significantly reduced. In sorghum plants, and low concentrations of Cd^2+^ (≤25 mg/kg) stimulated an increase in plant height, which may be associated with their certain resistance to Cd, but high levels of Cd^2+^ inhibited the height growth of plants of the sorghum genus [42]. Thus, it is possible that lower concentrations of Cd^2+^ under growing conditions can stimulate plant growth to a certain extent, while higher concentrations suppressed their growth [43].

Many studies show that PGPM treatment has a positive effect on plants, even in the presence of HMs in the medium. For example, alfalfa *Medicago sativa* L. plants treated with PGPR and grown in the presence of Cu, Pb, and Zn increased shoot length by 22–77% and shoot biomass by up to 220% compared to untreated plants [44,45]. Treatment of *Atriplex halimus* and *Arthrocnemum macrostachyum* plants growing in the presence of HM showed an improvement in their morphometric parameters compared to the untreated controls. This may be due to the fact that microorganisms improve plant nutrition by dissolving phosphates, iron, and nitrogen fixation; in addition, they can stimulate plant growth by secreting auxins [46,47,48,49,50,51].

Thus, plant resistance to the toxic effect of Cd may be due to more efficient root growth due to the positive effect of substances released by microorganisms and a decrease in the concentration and accumulation of HMs in the plant root system. The predominant accumulation of Cd in the roots compared to its accumulation in the aboveground plant organs is determined by the barrier functions of the plant root system in relation to toxic HMs [52]. In our opinion, the ability of bacteria to colonize the surface of the roots makes an important contribution to reducing the toxicity of Cd in the environment.

## 5. Conclusions

The present study showed that seed inoculation with *Pseudomonas sp.* OBA 2.4.1, isolated from the rhizosphere of the *Oxytropis baschkiriensis*, had a growth-promoting and protective effect on pea plants under Cd stress. Treatment with OBA 2.4.1 resulted in a 2.6-fold increase in seedling length compared to the untreated plants. The bacterium exhibits PGP properties, including biofilm formation and siderophore production. Moreover, the strain OBA 2.4.1 contributed to the reduction of oxidative stress caused by Cd. The level of hydrogen peroxide and MDA is significantly lower than in untreated stressed plants. In addition, bacteria are able to colonize the roots of pea plants. This ability of bacteria also positively affects the growth and development of plants under stress. The evaluation of the current study suggests that the strain *Pseudomonas* sp. OBA 2.4.1, can potentially be used as a promising alternative and an environmentally friendly approach to facilitating pea growth and stress tolerance under cadmium stress. However, further field experiments are required to evaluate its full potential for mitigating heavy metals-caused stress in plants.

## Figures and Tables

**Figure 1 biotech-12-00005-f001:**
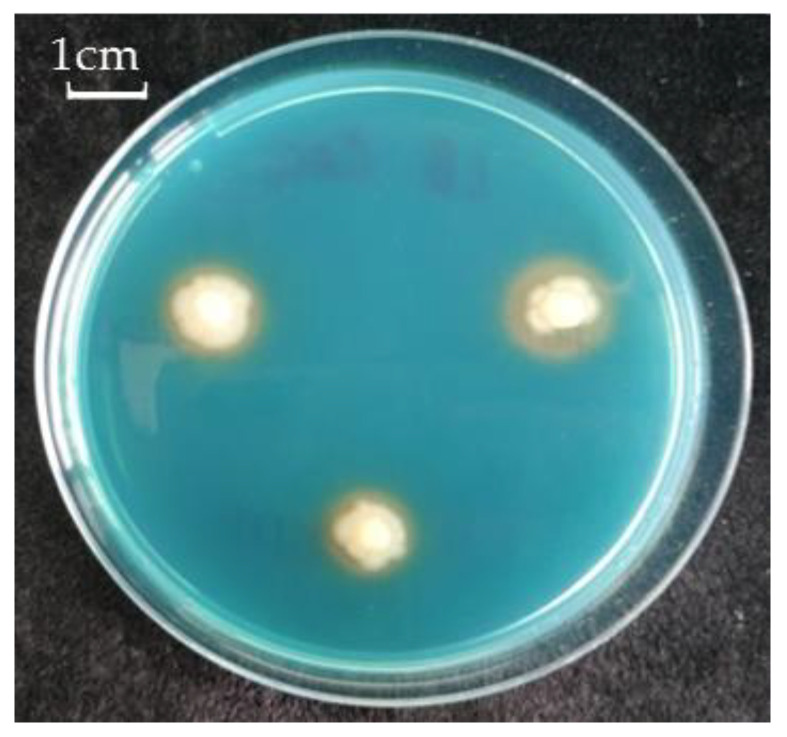
Siderophore production of *Pseudomonas* sp. OBA 2.4.1. The bar in the photograph indicates 1 cm.

**Figure 2 biotech-12-00005-f002:**
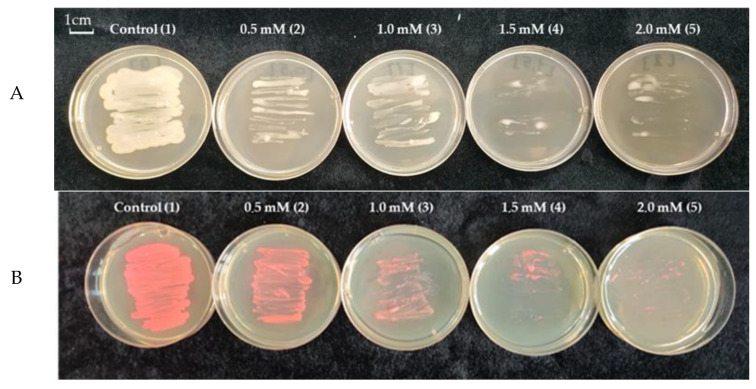
**A**—The growth of the strain *Pseudomonas* sp. OBA 2.4.1 (1) on LB medium, (2) with the addition of 0.5 mM Cd, (3) with the addition of 1 mM Cd, (4) with the addition of 1.5 mM Cd, (5) with the addition of 2 mM Cd; **B**—the growth of the strain *Pseudomonas* sp. OBA 2.4.1 transformed with plasmid pJNTurboRFP (which gives a pink color to the bacteria) (1) on LB medium, (2) with addition of 0.5 mM Cd, (3) with addition of 1 mM Cd, (4) with addition of 1.5 mM Cd, (5) with the addition of 2 mM Cd. The bar in the photograph indicates 1 cm.

**Figure 3 biotech-12-00005-f003:**
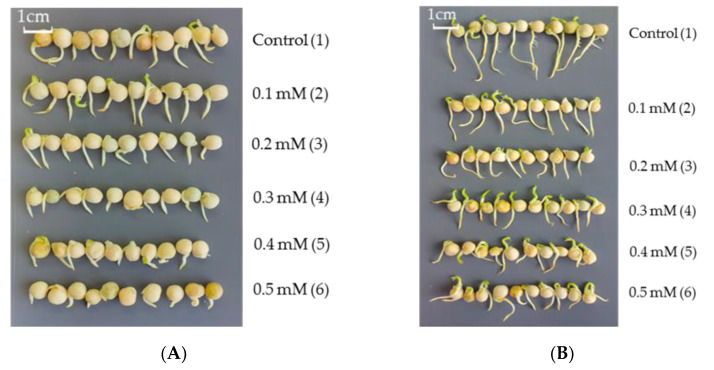
Effect of Cd on the growth of pea seeds germinated on filter paper in sterile water (control) and in the presence of Cd: 0.1, 0.2, 0.3, 0.4, 0.5 mM (**A**); the second variant of the seeds was treated with *Pseudomonas* sp. OBA 2.4.1 and also grew in the presence of Cd: 0.1, 0.2, 0.3, 0.4, 0.5 mM (**B**). The bar in the photograph indicates 1 cm.

**Figure 4 biotech-12-00005-f004:**
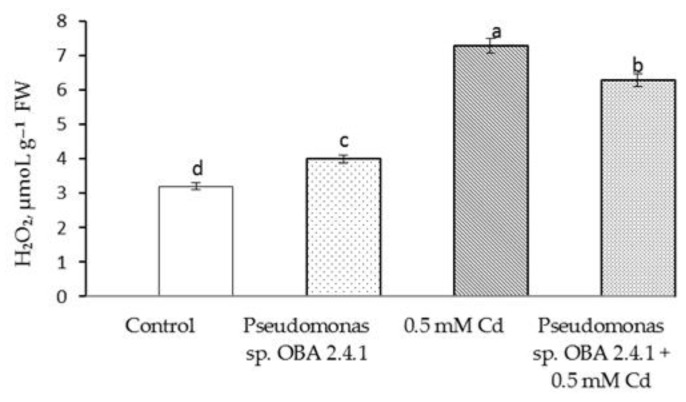
Effect of *Pseudomonas* sp. OBA 2.4.1 on the content of H_2_O_2_ in 7-day-old pea seedlings under normal and Cd stress conditions. The bars indicate the mean values of three replicates ± SEM. Different lowercase letters indicate a significant difference between the means at the level of P < 0.05 (ANOVA, LSD test).

**Figure 5 biotech-12-00005-f005:**
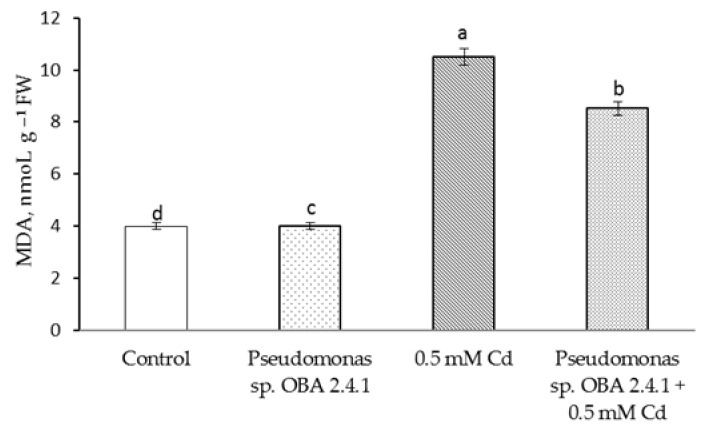
Effect of *Pseudomonas* sp. OBA 2.4.1 on the content of MDA in 7-day old pea seedlings under normal and Cd stress conditions. The bars indicate the mean values of three replicates ± SEM. Different lowercase letters indicate a significant difference between the means at the level of P < 0.05 (ANOVA, LSD test).

**Figure 6 biotech-12-00005-f006:**
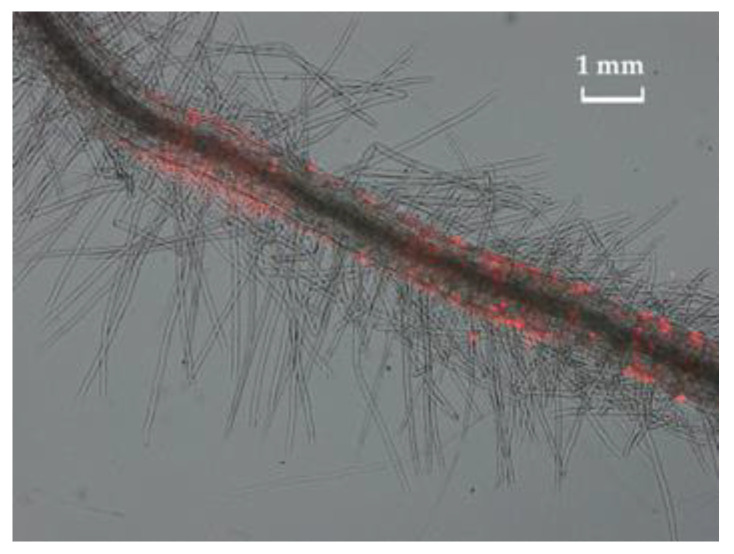
Bacteria transformed with the pJNTurboRFP vector on the surface of pea plant root hairs. Microcolonies of *Pseudomonas* sp. 2.4.1 formed after 1 day of incubation. Visualization was made using a fluorescent microscope Axio Imager M1 (Carl Zeiss, Jena, Germany). The bar in the photograph indicates 1 mm.

**Table 1 biotech-12-00005-t001:** Average length of seedlings of pea plants under different concentrations of cadmium. The presented data are the average of three repetitions (n = 50).

Concentrations of Cd, mM	Without inoculation, mm	Inoculation with *Pseudomonas* sp. OBA 2.4.1, mm
0 (control)	11.8 ± 0.35	30.9 ± 0.93
0.1	14.5 ± 0.43	23.5 ± 0.70
0.2	18.6 ± 0.55	22.4 ± 0.67
0.3	15.6 ± 0.46	21.3 ± 0.63
0.4	12.7 ± 0.38	19.2 ± 0.57
0.5	8.6 ± 0.25	18.8 ± 0.56

## Data Availability

Not applicable.

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
