# Peer review of "Effects of Pseudomonas sp. OBA 2.4.1 on Growth and Tolerance to Cadmium Stress in Pisum sativum L."

_biotech, 2023, doi:10.3390/biotech12010005_

Round 1

Reviewer 1 Report

Comments are in the attached file.

Author Response

Dear Reviewer we thank you for the attentive and benevolent attitude of our article.  We agree with your comments. Corrections made to the manuscript.

  1. English is very poor. Must revise the full abstract.

     The abstract has been reworked.

  1. spell out - SOD

    deciphered in the text - superoxide dismutase (SOD)

  1. add reference

     references is add

  1. Add web access link in ref section.

        web access link in ref section added [23].

  1. obtained by isolated microorganisms

          changed to bacteria

  1. Must include sequence submission information and NCBI accession no and elaborate analysis and BLAST report.

 Corrected The sequenced 16S rRNA sequence was deposited in GenBank (http://www.ncbi.nlm.nih.gov/genbank) under the number OK039351 and in the rpoD gene under the number OM641958. When comparing fragments of the 16S rRNA and rpoD gene sequences with typical strains, it was found that Pseudomonas sp. OBA 2.4.1 is closest in homology to Pseudomonas fluorescens. This strain is described in detail in a study by [28].

  1. "," replace with "."

Figures and  Table are replaced

Text revised, supplemented and corrected

With gratitude,

Dr. Liliya Khakimova  and co-authors

Reviewer 2 Report

The present article is based on an interesting theme. The authors have well designed the work but the quality of presentation and the  language of paper is not satisfactory. Author should extensively revised the english of article. I am mentioning herewith some points.

153- alcohol for 1 min and 10% sodium hypochlorite solution for 20 min-  I think this much higher concentration and duration of sodium hypochlorite may damage the plant.

Line-159-  Elaborate  germination energy

Line-175- and the sight ?

Line 184-187- reframe the sentence

Line-198- Influence of the strain Pseudomonas sp. OBA 2.4.1 on the growth and development of 198 pea plants was carried out. ?

Reframe line 277- OBA 2.4.1 showed good growth on medium with cadmium 277 stress at concentrations up to 1 mM

Author Response

Dear Reviewer we thank you for the attentive and benevolent attitude of our article.  We agree with your comments. Corrections made to the manuscript.

- We have been using this seed sterilization technique in our laboratory for many years. For example, references from the article [23, 28].

Line-159- corrected

Line-175-corrected

Line 184-187- corrected

Line-198 - corrected

 line 277- corrected

Reviewer 3 Report

The manuscript, Effects of Pseudomonas sp. OBA 2.4.1 on growth and tolerance to cadmium stress in Pisum sativum L. is a good study collated with suitable findings and data. However, it has several minor weaknesses. Manuscript can be accepted after minor revision.

·         Language must be checked across the manuscript

·         There are several flaws in sentences must be checked for the connectivity and flow of the content

·         Acronyms must be used accurately and uniformly

·         Avoid redundancy of the words

·         Language of the abstract needs to be improved. In just 8-9 lines of abstract,  Pseudomonas sp. OBA 2.4.1 has been repeated 4 times.

·         Treatment methods are not so clear in the methodology section. Bacterial tolerance of Cd is clear however, seed treatment / plant treatment is not so clearer. The author must write clearly and easy to understand

·         Why did author did not try to characterize the presence of czr gene cluster responsible for cadmium and zinc resistance (https://doi.org/10.1016/S0378-1119(99)00349-2 )

Minor comments:

·         Line 36-39: rephrase

·         Define acronym SOD on its first use

·         Line 46-49: provide citations

·         Use either Oryza sativa or rice, preferably Oryza sativa

·         Provide name of the country after Southern Urals.

·         Line 98-99: rephrase and change word computer analysis with computational analysis

·         Line 132: rephrase

·         Line 149-150: what was the wavelength used

·         Line 211: cadmium ions or CdCl2 , please maintain uniformity in using important words/key words that are often used in the manuscript

·         Kindly use decimal (.) in expressing the values not comma (,) for example 11,8±0,35 and  30,9±0,93

·         Kindly use either mM or µM in expressing the concentrations, it’s a bit confusing

·         Figure 6: if possible, try to compare the colonization and root hair growth with the control

Author Response

Dear Reviewer we thank you for the attentive and benevolent attitude of our article.  We agree with your comments. Corrections made to the manuscript.

Line 36-39: corrected

Line 46-49: corrected

Line 98-99: corrected

Line 132: corrected

Line 211 - corrected - Cd

 Figure 6: Not possible at the moment, but we will take note.

Thanks for the review, all edits have been made.

Reviewer 4 Report

Review report on the manuscript Effects of Pseudomonas sp. OBA 2.4.1 on growth and tolerance 2 to cadmium stress in Pisum sativum L.

The aim of this study was to determine the different activity and inhibitory concentrations of cadmium salts against the growth of Pseudomonas sp.  analysis of its effect on Pisum sativum L. under cadmium stress. The authors  showed that seed inoculation with Pseudomonas sp. OBA, isolated from the rhizosphere of the Oxytropis baschkiriensis, had a growth-promoting and protective effect on pea plants under Cd stress.

The manuscript is well written and the subject is of great interest. I have some minor comments.

1.       It is well known that  cadmium is a toxic metal that can cause serious health problems for humans, being necessary to find bioremediation solutions to reduce its harmful effects. There are several approaches used in order to annihilate toxicity induced by cadmium, as a solution for injury bioremediation in humans. In this context, the authors are asked to refer to some new published papers with relevance in this field, for example: https://doi.org/10.3390/ma14092257; https://doi.org/10.3390/app11094220; https://doi.org/10.1016/j.molstruc.2021.131325.

2.       More details are necessary regarding the isolation of Pseudomonas sp. in the section Materials and Methods.

3.       Details of microscopic images in Figure 1 A- scale, magnification should be added.

4.       The limitation of the study should be mentioned at the end of the discussion section.

Author Response

Dear Reviewer we thank you for the attentive and benevolent attitude of our article.  We agree with your comments. Corrections made to the manuscript.

  1. We have added a link https://www.mdpi.com/1996-1944/14/9/2257
  2. corrected
  3.  Figure is deleted
  4. corrected

Round 2

Reviewer 2 Report

The article can be accepted in the present form.